# Activation of GCN2 kinase by ribosome stalling links translation elongation with translation initiation

**Ryuta Ishimura[1†], Gabor Nagy[1†], Ivan Dotu[2], Jeffrey H Chuang[3,4], Susan L Ackerman[1,5,6]\***

[1]Howard Hughes Medical Institute, The Jackson Laboratory for Mammalian Genetics, Bar Harbor, United States; [2]Research Programme on Biomedical Informatics, Department of Experimental and Health Sciences, Universitat Pompeu Fabra, Barcelona, Spain; [3]The Jackson Laboratory for Genomic Medicine, Farmington, United States; [4]Department of Genetics and Genome Sciences, University of Connecticut Health Center, Farmington, United States; [5]Department of Cell and Molecular Medicine, University of California, San Diego School of Medicine, La Jolla, United States; [6]Section of Neurobiology, University of California, La Jolla, United States

**\*For correspondence:**
sackerman@ucsd.edu

[†]These authors contributed equally to this work

**Abstract** Ribosome stalling during translation has recently been shown to cause neurodegeneration, yet the signaling pathways triggered by stalled elongation complexes are unknown. To investigate these pathways we analyzed the brain of C57BL/6J-*Gtpbp2^nmf205-/-* mice in which neuronal elongation complexes are stalled at AGA codons due to deficiencies in a tRNA$^{Arg}_{UCU}$ tRNA and GTPBP2, a mammalian ribosome rescue factor. Increased levels of phosphorylation of eIF2α (Ser51) were detected prior to neurodegeneration in these mice and transcriptome analysis demonstrated activation of ATF4, a key transcription factor in the integrated stress response (ISR) pathway. Genetic experiments showed that this pathway was activated by the eIF2α kinase, GCN2, in an apparent deacylated tRNA-independent fashion. Further we found that the ISR attenuates neurodegeneration in C57BL/6J-*Gtpbp2^nmf205-/-* mice, underscoring the importance of cellular and stress context on the outcome of activation of this pathway. These results demonstrate the critical interplay between translation elongation and initiation in regulating neuron survival during cellular stress.

## Introduction

Protein translation cycles through the phases of initiation, elongation, termination, and ribosome recycling. Translation has been thought to be predominantly regulated at the initiation step in which the 40S small ribosome subunit, preloaded with Met-tRNAi, binds to the 5' end of mRNA. This complex then scans the mRNA for the AUG initiation codon where the 60S large ribosome subunit joins to form the 80S monosome (*Aitken and Lorsch, 2012*; *Hinnebusch, 2014*). A major control point of initiation is the reversible phosphorylation of the eukaryotic initiation factor 2 (eIF2), which in its active form binds GTP and Met-tRNA$_i^{Met}$ and delivers the initiator tRNA to the 40S subunit of the ribosome (*Baird and Wek, 2012*; *Donnelly et al., 2013*). A variety of stress responses regulate translation initiation by converging on phosphorylation of Ser51 of the α subunit of eIF2, which prevents formation of the ternary complex and thus translation initiation (*Dalton et al., 2012*). In addition to globally repressing translation, phosphorylation of eIF2α also enhances translation of specific mRNAs with open reading frames in their 5' leaders, such as ATF4 (*Lu et al., 2004*; *Vattem and*

**eLife digest** Information stored in DNA is used to make proteins in a two-step process. First, the DNA is copied to make molecules of messenger ribonucleic acid (or messenger RNA for short). Next, machines called ribosomes use the messenger RNAs as templates to assemble chains of amino acids – the building blocks of proteins – in a process called translation. Another type of RNA molecule called transfer RNA carries each amino acid to the ribosomes. If a specific transfer RNA is not available for translation at the right time, the ribosome might stall as it moves along the messenger RNA. At this point, the ribosome needs to be restarted or it will fall off the mRNA without finishing the protein.

In 2014, a group of researchers reported that certain types of brain cells are very sensitive to ribosome stalling, and tend to die if translation does not continue. A protein called GTPBP2 was shown to play an important role in restarting stalled ribosomes in these cells. Here, Ishimura, Nagy et al. – including some of the researchers from the earlier work – investigated the molecular pathways that ribosome stalling triggers in brain cells using mutant mice that lacked the GTPBP2 protein. The experiments show that ribosome stalling activates an enzyme known as GCN2, which was already known to sense other types of malfunctions in cellular processes.

Ishimura, Nagy et al. also show that GCN2 triggers stress responses in the cells by activating a communication system called the ATF4 pathway. This pathway protects the cells from damage, and its absence results in more rapid cell deterioration and death. The next challenges are to understand the exact mechanism by which GCN2 senses stalled ribosomes, and to find out how ribosome stalling causes the death of brain cells.

---

*Wek, 2004*). This transcription factor promotes transcription of genes involved in a wide range of adaptive functions including amino acid metabolism, redox control, and translational control (*Harding et al., 2003*; *Huang et al., 2009*; *Kilberg et al., 2012*).

The eIF2α subunit is the common target of four protein kinases, the activation of which is induced by distinct and specific extracellular and intracellular stresses. PKR-like ER kinase (PERK, EIF2AK3) is generally activated by the accumulation of misfolded proteins in the endoplasmic reticulum (*Ron and Harding, 2012*; *Zhao and Ackerman, 2006*). Heme-regulated inhibitor (HRI, EIF2AK1) and protein kinase R (PKR, EIF2AK2) are activated in erythroid cells by heme deprivation and viral infections, respectively (*Chen, 2014*; *Marchal et al., 2014*). Finally, GCN2 (General Control Non-derepressible 2, EIF2AK4) is activated by low intracellular levels of amino acids (*Castilho et al., 2014*). Amino acid deficiency results in an increase of deacylated tRNA species, which bind the histidyl-tRNA synthetase-related domain of GCN2 and the C-terminus, domains that are essential for GCN2 activation (*Dong et al., 2000*; *Wek et al., 1995*; *Zhu et al., 1996*).

Elongation has recently emerged as a translation phase that is also subject to elaborate regulatory and surveillance mechanisms (*Richter and Coller, 2015*). During elongation the ribosome moves in a codon-dependent fashion down the mRNA while amino acids are cyclically presented to the A site of ribosomes by EF-1/EF-T-bound aminoacylated tRNAs and added to the nascent peptide chain by the formation of peptide bonds. The rate of elongation is a function of ribosome transit rate, i.e. the time the translating ribosome spends on each codon, which is influenced by the secondary structure of the mRNA, nascent peptide:ribosome interactions, as well as codon identity and the availability of properly modified, cognate tRNAs for these codons (*Chen et al., 2013*; *Ishimura et al., 2014*; *Ito and Chiba, 2013*; *Koutmou et al., 2015*; *Lu and Deutsch, 2008*; *Nedialkova and Leidel, 2015*; *Quax et al., 2015*; *Wen et al., 2008*). Elongation rates are now considered important factors in gene expression that determine the fate of mRNA undergoing translation and the functioning of the nascent proteins (*Elvekrog and Walter, 2015*; *Inada, 2013*; *Pelechano et al., 2015*; *Presnyak et al., 2015*; *Quax et al., 2015*; *Sherman and Qian, 2013*). Pausing of translating ribosomes can be induced by cellular stress including temperature and oxidative stress (*Knight et al., 2015*; *Liu et al., 2013*; *Shalgi et al., 2013*; *Zhong et al., 2015*). Furthermore, misregulation of elongation may contribute to disease pathology resulting in neurological

dysfunction such as observed in Fragile X syndrome (*Chen et al., 2014*; *Darnell et al., 2011*; *Udagawa et al., 2013*).

Recently we reported that ribosome stalling is intimately associated with cerebellar and retinal degeneration in the C57BL6/J (B6J)-*nmf205* mutant mouse (*Ishimura et al., 2014*). In this mouse model, two mutations are necessary to induce neuron loss: a loss of function mutation (*Gtpbp2$^{nmf205-/-}$*) in a translational GTPase GTPBP2 (guanosine triphosphate-binding protein 2) and a hypomorphic mutation in the nervous system-specific n-Tr20 (nTRtct5) tRNA$^{Arg}_{UCU}$ gene. Ribosomal profiling of cerebella from mice with the n-Tr20 mutation revealed an increase in ribosome occupancy at AGA codons that dramatically increased in the absence of GTPBP2. These studies demonstrated that GTPBP2 likely functions as a ribosome rescue factor and that ribosome stalling, i.e. abnormally long ribosome transit rates, could lead to neuron death. Recently, a mutation in the *GTPBP2* gene was suggested to underlie cerebellar and retinal degeneration and intellectual disability in humans, supporting an essential role of GTPBP2 in neuronal homeostasis (*Jaberi et al., 2015*).

Here, we take advantage of unique mouse models with the *Gtpbp2$^{nmf205-/-}$* mutation to assess the signaling events that are triggered by ribosome stalling. We show that GCN2 is activated in the B6J-*Gtpbp2$^{nmf205-/-}$* mutant brain in a tRNA$^{deacyl}$-independent manner prior to the onset of neurodegeneration. Activation of this kinase results in eIF2α phosphorylation and induction of ATF4-target genes in the cerebellum and hippocampus of B6J-*Gtpbp2$^{nmf205-/-}$* mice. Further, our genetic studies demonstrate that GCN2 activation functions to attenuate neurodegeneration in the B6J-*Gtpbp2$^{nmf205-/-}$* brain. These results suggest that ribosome stalling activates GCN2 as a previously unknown regulatory mechanism that links the elongation and initiation steps of translation.

## Results

### ATF4 target genes are induced in the B6J-*Gtpbp2$^{nmf205-/-}$* brain

To begin to elucidate the molecular pathways activated by ribosome stalling, we performed gene expression studies on cerebella isolated from B6J (n-Tr20, also known as *nTRtct5*, mutant) and B6J-*Gtpbp2$^{nmf205-/-}$* (n-Tr20 mutant; *Gtpbp2$^{nmf205-/-}$* ) mice at 5 weeks of age, a time when many granule cells are undergoing apoptosis in mutant mice (*Figure 1A* and *Figure 1—figure supplement 1*). To separate changes in gene expression caused by ribosome stalling from more general changes caused by neuron damage, gene expression analysis was also performed on the hippocampus of 5-week-old mutant mice (which has low numbers of apoptotic neurons relative to the mutant cerebellum) and from both brain regions in 3-week-old mice (a time when no neuron death is observed) (*Figure 1A* and *Figure 1—figure supplements 1* and *2*). Gene Chip expression arrays (Affymetrix Mouse Gene 1.0 ST Arrays) were performed with three biological replicates for each age, genotype, and tissue. Genes whose expression differed between B6J and B6J-*Gtpbp2$^{nmf205-/-}$* by at least 1.5 fold with a q-value < 0.05 were selected for further analysis (*Figure 1—figure supplement 3* and *Figure 1—source data 1*). Using these criteria, 583 genes were upregulated and 327 genes were downregulated in the cerebellum from 5-week-old mutant mice. Fewer genes in the hippocampus of 5-week-old mutant mice exhibited altered expression (78, upregulated; 26, downregulated). In the cerebellum of 3-week-old B6J-*Gtpbp2$^{nmf205-/-}$* mice 225 genes were upregulated and 100 downregulated, while only 10 genes were upregulated and 1 downregulated in the hippocampus of mutant mice at this age.

Neuron damage induces activation of microglia and astrocytes and these inflammatory reactions are commonly observed in acute and chronic neurodegeneration. Indeed, Kegg pathway analysis and upstream regulator pathway analysis (Ingenuity Pathway Analysis) on differentially expressed genes in the B6J-*Gtpbp2$^{nmf205-/-}$* brain revealed enrichment of inflammation/immune pathways (e.g., hematopoietic cell lineage and cytokine-cytokine receptor interactions) (*Figure 1B,C* and *Figure 1—source data 2*). We compared our differentially regulated gene set to genes that were activated in microglia and/or astrocytes in the spinal cord of transgenic mice for mutant G93A SOD1, a model of amyotrophic lateral sclerosis (*Chiu et al., 2013*), or the cortex of mice transgenic for both the APP$^{Swe}$ and PS1dE9 genes, a commonly used Alzheimer's Disease model (*Orre et al., 2014*) (Glia Open Access Database) (*Holtman et al., 2015*). Overlap in our gene set was observed with genes expressed in activated microglia or astrocytes (150 and 60 genes, respectively) or both cell types (39 genes) (*Figure 1D* and *Figure 1—source data 3*). Upregulation of these genes was highest in the 5-

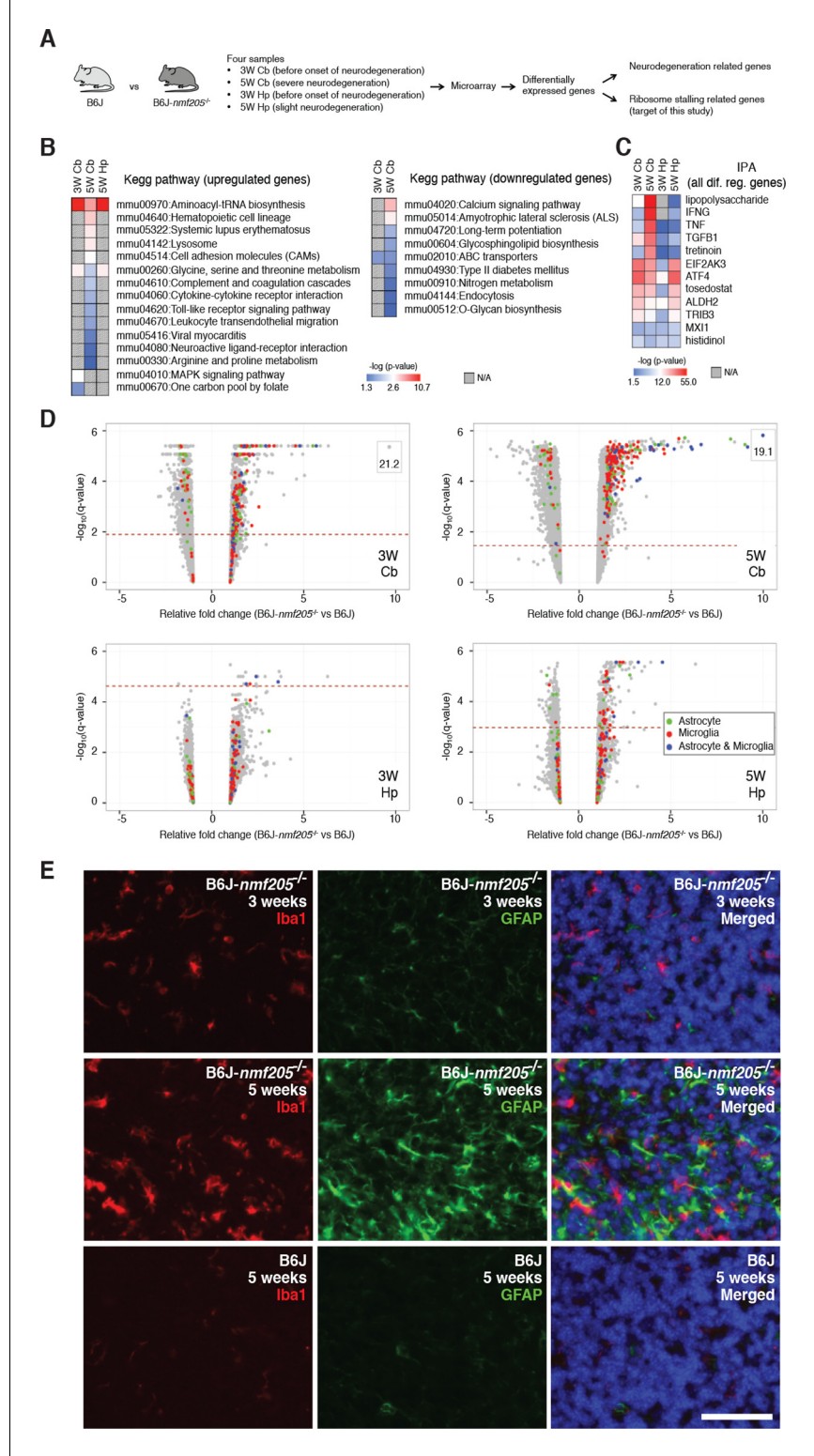

**Figure 1.** Transcriptional profiling of B6J-*Gtpbp2*[nmf205-/-] mutant mice. (**A**) To segregate genes associated with ribosome stalling from genes nonspecifically induced during neurodegeneration, microarray analysis was performed on the cerebellum (Cb) and hippocampus (Hp) from B6J and B6J-*Gtpbp2*[nmf205-/-] (B6J-*nmf205*[-/-]) mice at 3-weeks (3W) and 5-weeks (5W) of age. (**B**) Enriched pathways among genes that were up or downregulated in the B6J-*Gtpbp2*[nmf205-/-] brain were obtained by Kegg pathway analysis. Note that upregulated gene pathways exhibited more robust p-values. (**C**) Ingenuity Pathway Analysis (IPA) upstream regulator analysis of B6J-

*Figure 1 continued on next page*

*Figure 1 continued*

*Gtpbp2^{nmf205-/-}* upregulated genes demonstrates ATF4 and EIF2AK3 activation in the 3- and 5-week hippocampus and cerebellum. Inflammatory-related upstream regulators are also strongly activated of in the 5-week-old mutant cerebellum. (D) Volcano plots of $-\log_{10}$q-values and relative fold changes of all genes. Genes previously shown to be expressed in microglia (red), astrocytes (green), or in both microglia and astrocytes (blue) are indicated, with all other genes shown in gray. Genes with very high levels of induction are shown in boxes to fit on the scale. The horizontal dashed lines indicate a false discovery rate of 0.1. Note the strong induction of genes activated in both microglia and astrocytes in the 5-week-old B6-*Gtpbp2^{nmf205-/-}* cerebellum. (E) Immunofluorescence using antibodies to Iba1 (microglia) and GFAP (astrocytes) was performed on sections of B6J and B6J-*Gtpbp2^{nmf205-/-}* cerebellum at indicated ages. Microglia and astrocytes were activated in the mutant cerebellum with highest levels of activation at 5 weeks of age. Sections were counterstained with Hoechst 33342 to visualize nuclei as shown in merged. Scale bar, 50 μm.

The following source data and figure supplements are available for figure 1:

**Source data 1.** Significantly changed genes.

**Source data 2.** GO & IPA analysis.

**Source data 3.** Inflammatory genes.

**Figure supplement 1.** Progressive neurodegeneration in B6J-*Gtpbp2^{nmf205-/-}* mice.

**Figure supplement 2.** Neurodegeneration is reduced in hippocampus of B6J-*Gtpbp2^{nmf205-/-}* mice.

**Figure supplement 3.** Microarray analysis of the B6J and B6J-*Gtpbp2^{nmf205-/-}* cerebellum.

**Figure supplement 4.** The inflammatory response in the B6J-*Gtpbp2^{nmf205-/-}* hippocampus.

week-old B6J-*Gtpbp2^{nmf205-/-}* cerebellum consistent with the robust signal from immunofluorescence on mutant and wild type brains with antibodies to Iba1 and GFAP, markers for activated microglial and astrocytes, respectively (*Figure 1E* and *Figure 1—figure supplement 4*).

In addition to upregulation of inflammation/immune genes, G̲ene O̲ntology and Kegg analysis of upregulated genes identified enrichment of genes in aminoacyl-tRNA biosynthesis and amino acid metabolic pathways in the mutant cerebellum and hippocampus from 3- and 5-week-old mice (*Figure 1B* and *Figure 1—source data 2*). Consistent with this result, upstream regulator analysis of differentially expressed genes in the mutant brain demonstrated enrichment for EIF2AK3 (PERK), an eIF2α kinase, and its downstream effector, activating transcription factor 4 (ATF4), a key component of the integrated stress response and activator of aminoacyl-tRNA synthetases and amino acid metabolic pathways (*Figure 1C* and *Figure 1—source data 2*). P-values for pathways that were enriched in downregulated genes in the mutant brain tended to be lower, but were consistent with dysfunction of terminally differentiated neurons (e.g., calcium signaling and amyotrophic lateral sclerosis pathways).

To further investigate ATF4 activation in the B6J-*Gtpbp2^{nmf205-/-}* cerebellum we compared our differentially expressed gene sets to 472 ATF4 target genes identified by chromatin immunoprecipitation sequencing (ChIP-seq) from tunicamycin-treated mouse embryonic fibroblasts (*Han et al., 2013*). Ninety-nine of these known target genes were upregulated in the cerebellum and/or hippocampus of mutant mice (*Figure 2A* and *Figure 2—source data 1*). ATF4 target genes were upregulated in both the cerebellum and hippocampus; of the 56 genes that were upregulated in both the 3- and 5-week mutant cerebellum and in the 5-week mutant hippocampus, 73% of genes were known ATF4 target genes (*Figure 2B* and *Figure 2—source data 2*). Although fewer genes were significantly upregulated in the 3-week mutant hippocampus, 6 of 10 of these genes were known ATF4 targets (*Figure 2C*). ATF4 target genes were among the most highly upregulated genes in the cerebellum and hippocampus (*Figure 2D* and *Figure 2—source data 1*). Together, these data suggested that ATF4 activation is an early and robust event in both the hippocampus and cerebellum of B6J-*Gtpbp2^{nmf205-/-}* mice.

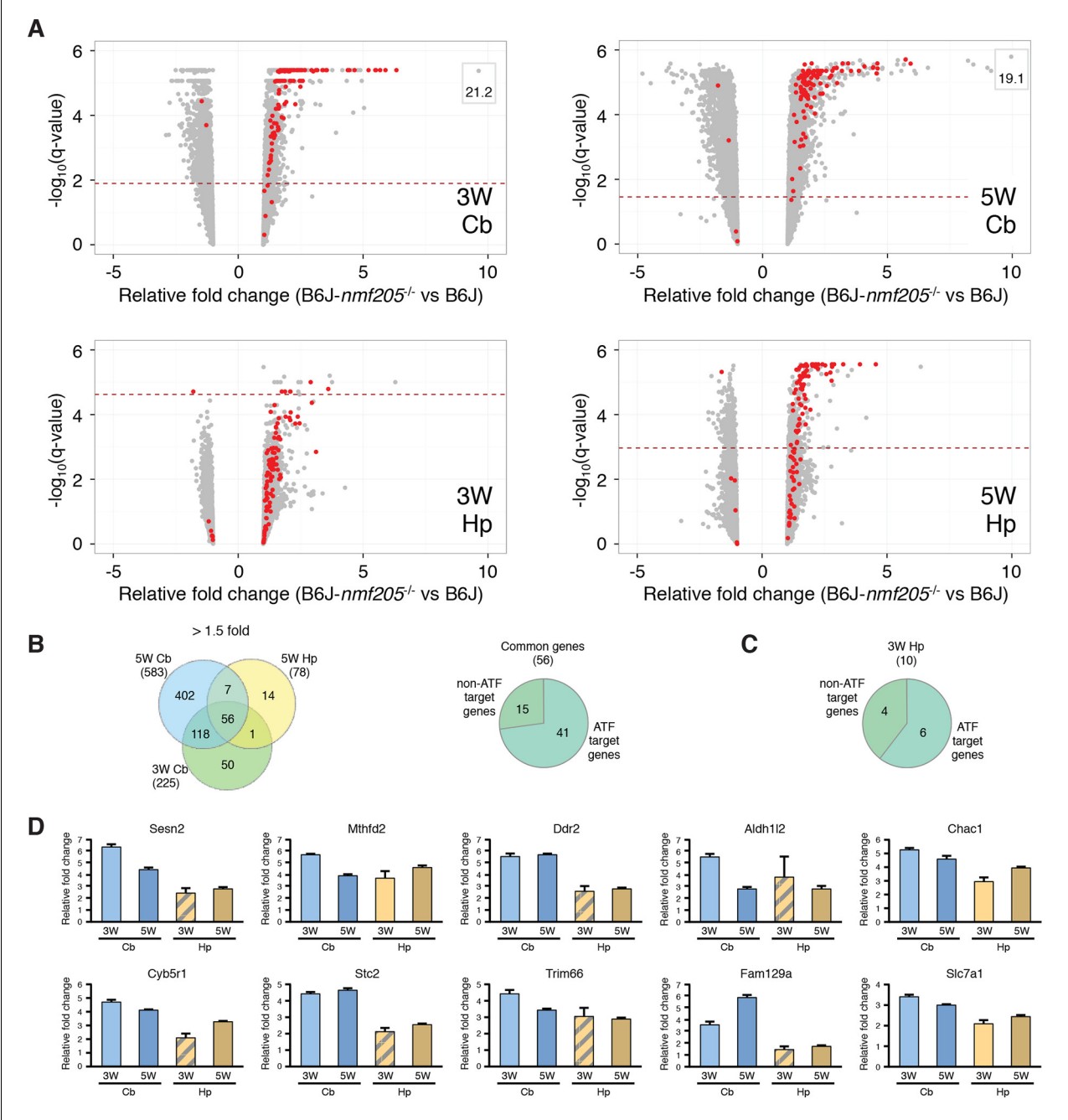

**Figure 2.** ATF4 downstream genes are robustly activated in B6J-*Gtpbp2*[nmf205-/-] mutant brain. (A) Volcano plots of ATF4 downstream genes demonstrating activation of the ATF4 pathway in the B6J- *Gtpbp2*[nmf205-/-] (B6J-*nmf205*[-/-]) brain. Fold change and q values of all genes are shown and ATF4 target genes, obtained from previous ChIP-Seq studies as described in the text, are red. Genes with very high levels of induction are shown in boxes to fit on the scale. Note that although induction of ATF4 downstream genes is observed in all groups, upregulation of these genes occurs earlier and is most robust in the cerebellum. (B) Venn diagram of differentially expressed genes between B6J and B6J-*Gtpbp2*[nmf205-/-] 3-and 5-week cerebellum and 5-week hippocampus. Note that a large number of genes (56 genes) are consistently upregulated between cerebellum and hippocampus and that most of these genes are known ATF4 target genes. (C) The distribution of *ATF4* target genes in the upregulated gene set from the B6J-*Gtpbp2*[nmf205-/-] 3-week hippocampus. (D) The relative fold change (RFC) of the top ten upregulated genes in the 3-week-old B6J-*Gtpbp2*[nmf205-/-] cerebellum for each time point and brain region. RFCs depicted with solid bars were significant changes (p<0.05) and those depicted by hatched bars were not.

The following source data is available for figure 2:

*Figure 2 continued on next page*

*Figure 2 continued*

**Source data 1.** Differentially expressed ATF4 target genes.
**Source data 2.** Common differentially expressed genes.

## Activation of the eIF2α kinase GCN2 in the B6J-*Gtpbp2*$^{nmf205-/-}$ brain

Preferential translation of ATF4 and activation of the integrated stress response is initiated by phosphorylation of Ser51 of eIF2α. Thus, we examined the eIF2α phosphorylation status in the cerebellar extracts of 3-week-old B6J, B6J.B6N$^{n-Tr20}$ (B6J mice in which the wild-type *n-Tr20* gene was transferred to B6J by repeated backcrossing), and B6J-*Gtpbp2*$^{nmf205-/-}$ mice by Western blot analysis (*Figure 3A*). Consistent with our ribosome occupancy studies at AGA codons (*Ishimura et al., 2014*), the ratio of p-eIF2α to total eIF2α was significantly higher in the B6J cerebellum relative to the B6J.B6N$^{n-Tr20}$ cerebellum, and the level of phosphorylated eIF2α further increased in the cerebellum of B6J-*Gtpbp2*$^{nmf205-/-}$ mice. Levels of total eIF2α did not differ significantly between genotypes.

Pathway analysis suggested activation of PERK, the kinase coupling eIF2α phosphorylation and ATF4 translation during the unfolded protein response (UPR). To determine if the UPR is activated in the B6J-*Gtpbp2*$^{nmf205-/-}$ cerebellum, we examined transcripts of *Hspa5*, also known as *BiP/GRP78*, and *Xbp1*, genes that are sensitive markers of the UPR and not under ATF4 regulation (*Kozutsumi et al., 1988*; *Yoshida et al., 2001*). During UPR activation, an intron in *Xbp1* is removed by the serine/threonine kinase/endoribonuclease, IRE1. This splicing, and activation of the ATF6 transcription factor lead to transcriptional upregulation of the ER chaperone BiP. Neither *Xbp1* splicing nor *BiP* levels were upregulated in the B6J-*Gtpbp2*$^{nmf205-/-}$ cerebellum (*Figure 3—figure supplement 1*, and data not shown) suggesting eIF2α phosphorylation and ATF4 activation was regulated by a eIF2α kinase other than PERK.

The eIF2α kinase GCN2 is activated by amino acid deprivation and other conditions that increase levels of uncharged tRNAs. Amino acid deprivation would be expected to also result in ribosome stalling. Thus we hypothesized that GCN2 is the primary effector of eIF2α phosphorylation in the B6J-*Gtpbp2*$^{nmf205-/-}$ brain. To test this, B6J.*Gcn2*$^{-/-}$ and B6J-*Gtpbp2*$^{nmf205-/-}$ mice were crossed to generate B6J-*Gtpbp2*$^{nmf205-/-}$; *Gcn2*$^{-/-}$ mice and phosphorylation of eIF2α in cerebellar extracts from these mice and B6J.*Gcn2*$^{-/-}$ mice was examined. Levels of p-eIF2α (Ser51) in the cerebellum from B6J-*Gtpbp2*$^{nmf205-/-}$; *Gcn2*$^{-/-}$ mice were similar to those observed in the cerebellum of B6J.B6N$^{n-Tr20}$ and B6J.*Gcn2*$^{-/-}$ mice demonstrating that GCN2 is the kinase responsible for the increase in p-eIF2α levels (*Figure 3A*).

To determine if loss of *Gcn2* also impacts ATF4 activation in the B6J-*Gtpbp2*$^{nmf205-/-}$ cerebellum, we first analyzed several known ATF4-target genes that were highly induced in the B6J-*Gtpbp2*$^{nmf205-/-}$ cerebellum. RT-qPCR was performed on cerebellar cDNA generated from 3-week-old B6J, B6J-*Gtpbp2*$^{nmf205-/-}$, B6J.*Gcn2*$^{-/-}$, and B6J-*Gtpbp2*$^{nmf205-/-}$; *Gcn2*$^{-/-}$ mice (*Figure 3—figure supplement 2*). As expected, we observed significant induction of these genes in B6J-*Gtpbp2*$^{nmf205-/-}$ relative to levels in the B6J cerebellum. However, in the absence of GCN2, upregulation of these genes was dramatically attenuated suggesting ATF4 activation was also regulated by GCN2 in the B6J-*Gtpbp2*$^{nmf205-/-}$ cerebellum.

To further define GCN2-dependent changes in gene expression, we performed RNA-Seq analysis from cerebella isolated from 3-week-old B6J, B6J-*Gtpbp2*$^{nmf205-/-}$, B6J.*Gcn2*$^{-/-}$, and B6J-*Gtpbp2*$^{nmf205-/-}$; *Gcn2*$^{-/-}$ mice. Three biological replicates were analyzed for each genotype and genes with expression differences $\geq$ 1.5 fold (padj < 0.01) were chosen for further analysis (*Figure 3—figure supplement 3*). Significant upregulation of 146 genes, many (57%) of which were also found to be upregulated by our microarray studies, was observed in B6J-*Gtpbp2*$^{nmf205-/-}$ cerebellum relative to the B6J cerebellum (*Figure 3B*). RNA-Seq also identified 191 genes that were significantly downregulated in B6J-*Gtpbp2*$^{nmf205-/-}$ mice, many (45%) of which were also identified by microarray (*Figure 3B*).

Loss of GCN2 reverted the expression of 83 of the 146 genes upregulated in the B6J-*Gtpbp2*$^{nmf205-/-}$ cerebellum to B6J levels, or near-B6J levels, including those genes with the highest fold induction in the B6J-*Gtpbp2*$^{nmf205-/-}$ cerebellum (*Figures 3B*, *Figure 3—source data 1*). Kegg

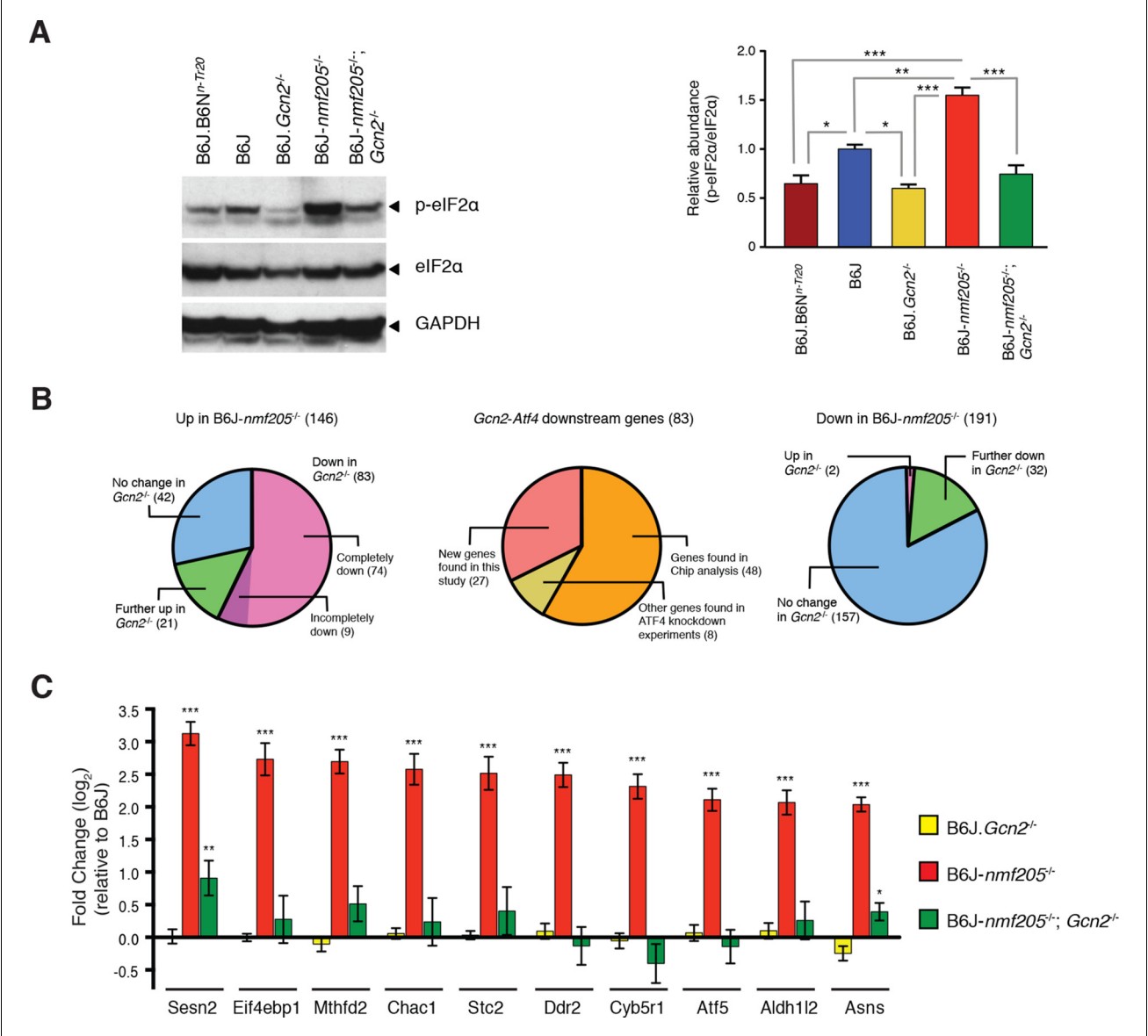

**Figure 3.** GCN2 mediates eIF2α phosphorylation and ATF4 activation in the B6J-*Gtpbp2*$^{nmf205-/-}$ brain. (**A**) Western blot analysis of cerebellar extracts from 3-week-old mice incubated with the antibodies shown (left panel). The relative abundance of phospho-eIF2α (Ser51) to total eIF2α was averaged for three biological replicates (right panel). Values are shown relative to B6J. (**B**) The effect of *Gcn2* deletion on the 146 genes upregulated or the 191 genes downregulated by 1.5 fold (padj < 0.01) in the cerebellum of B6J-*Gtpbp2*$^{nmf205-/-}$ (*B6J-nmf205*$^{-/-}$) relative to that of B6J (left and right diagram, respectively). ATF4 target status of the 83 genes that are upregulated in the B6J-*Gtpbp2*$^{nmf205-/-}$ cerebellum relative to B6J and that also have a significant decrease in expression in the B6J-*Gtpbp2*$^{nmf205-/-}$; *Gcn2*$^{-/-}$ cerebellum relative to B6J- *Gtpbp2*$^{nmf205-/-}$ (middle diagram). (**C**) Fold change of upregulated ATF4 downstream genes from B6J-*Gcn2*$^{-/-}$, B6J-*Gtpbp2*$^{nmf205-/-}$, and B6J-*Gtpbp2*$^{nmf205-/-}$; *Gcn2*$^{-/-}$ mice relative to B6J expression. The 10 most upregulated genes in cerebellum of B6J-*Gtpbp2*$^{nmf205-/-}$ mice relative to that B6J are shown. Note that the increased expression of these GCN2-ATF4 downstream genes in the B6J-*Gtpbp2*$^{nmf205-/-}$ cerebellum was completely suppressed by deletion of *Gcn2*. Error bars = SEM. *p<0.05, **p<0.01, and ***p<0.001 (one-way ANOVA, A, C).

The following source data and figure supplements are available for figure 3:

**Source data 1.** GCN2 RNA-Seq.
**Source data 2.** GCN2-regulated ATF4 target genes.
**Source data 3.** GCN2 RNA-Seq GO & KEGG analysis.

*Figure 3 continued on next page*

*Figure 3 continued*

**Figure supplement 1.** Absence of changes in *Xbp1* splicing in the B6J-*Gtpbp2^nmf205-/-* brain.
**Figure supplement 2.** Quantitative RT-PCR showing the induction of *ATF4* downstream genes is under the control of GCN2.
**Figure supplement 3.** RNA-Seq analysis.
**Figure supplement 4.** Deletion of *Gcn2* decreases expression of *ATF4* target genes in the B6J cerebellum.

pathway analysis of the 83 genes that reverted to wild type levels revealed enrichment for aminoacyl tRNA biosynthesis. IPA analysis also yielded several upstream regulators indicative of ATF4 activation including PERK and GCN2, compounds that induce the UPR (thapsigargin, tunicamycin) or cause amino acid deprivation (histidinol) (*Figure 3—source data 2*). Of these 83 genes, 48 genes had been previously shown to be ATF4 target genes by ChIP-Seq analysis (*Figure 3B* and *Figure 3—source data 2*) (*Han et al., 2013*). Eight additional genes had been previously shown to be downregulated in *Atf4^-/-* cells suggesting that they may also be under ATF4 control (*Ebert et al., 2012*; *Harding et al., 2003*; *Lange et al., 2008*). Analysis of the remaining 27 genes, which may be direct or indirect targets of ATF4 genes, failed to reveal enriched pathways, perhaps due to the small sample size (*Figure 3B*, and *Figure 3—source data 2*). Of the genes that were downregulated in the B6J-*Gtpbp2^nmf205-/-* cerebellum relative to B6J, the majority were not affected by *Gcn2* loss, and Kegg pathway analysis of those that did change also failed to reveal enriched pathways (*Figure 3B*, *Figure 3—source data 3*).

In the *Gtpbp2^nmf205-/-*; B6J.*Gcn2^-/-*; cerebellum, we also observed a further increase in expression of 21 genes that were upregulated in the B6J-*Gtpbp2^nmf205-/-* cerebellum. GO and Kegg Pathway analysis revealed this gene set was enriched in inflammatory response genes and also contained genes targeted to the nucleolus that were involved in ribosome and ribonucleoprotein biogenesis (*Figure 3—source data 3*). Similarly, nucleolar genes, RNA processing genes and ribosome biogenesis genes were enriched in the 580 genes that were upregulated in the B6J-*Gtpbp2^nmf205-/-*; *Gcn2^-/-* cerebellum, but not in the B6J-*Gtpbp2^nmf205-/-* cerebellum. A number (547) of genes were also uniquely downregulated in the B6J-*Gtpbp2^nmf205-/-*; *Gcn2^-/-* cerebellum. GO analysis of this gene set revealed moderate enrichment for neuronal and synaptic genes. Together these data suggest that in addition to attenuating the ISR, loss of *Gcn2* in the B6J-*Gtpbp2^nmf205-/-* cerebellum results in upregulation of genes involved in nucleolar function and downregulation of neuron specific genes suggestive of nucleolar dysfunction and neuronal damage, respectively.

Although there was a difference in levels of p-eIF2$\alpha$, our RNA-Seq experiments showed no significant changes in gene expression between the cerebellum of B6J and B6J.*Gcn2^-/-* mice (*Figure 3—source data 1*). We suspected these changes may be subtle and below the detection limits of these experiments. To test this idea, we used RT-qPCR to determine the relative expression of the top 10 ATF4 targets that were upregulated in the B6J-*Gtpbp2^nmf205-/-* cerebellum in the cerebellum of B6J, B6J.B6N^n-Tr20, and B6J.*Gcn2^-/-* mice (*Figure 3—figure supplement 4*). Indeed, we found that six of these genes were upregulated in the B6J cerebellum relative to expression levels in the B6J.B6N^n-Tr20 cerebellum, demonstrating that loss of *n-Tr20* function leads to minor changes in the expression of ATF4-target genes. In agreement with our results from analysis of B6-*Gtpbp2^nmf205-/-* mice, deletion of *Gcn2* reduced the expression of these genes back to B6J.B6N^n-Tr20 levels.

Activation of GCN2 under conditions of nutrient deficiency has been attributed in part to binding of uncharged tRNAs to a histidyl-tRNA synthetase (HisRS)-like domain in the carboxyl (C)-terminus. Ribosome footprinting of the 3-week-old B6J-*Gtpbp2^nmf205-/-* cerebellum revealed ribosome stalling only at AGA codons, as predicted by the decrease in processed *n-Tr20* (*Ishimura et al., 2014*). Levels of tRNA$^{Arg}_{UCU}$ in the cerebellum of 3-week-old B6J-*Gtpbp2^nmf205-/-* were 40% lower than in B6J mice (*Figure 4A*; n=3, p<0.05). However, the ratio of uncharged to charged tRNA$^{Arg}_{UCU}$ did not differ between the B6J and B6J-*Gtpbp2^nmf205-/-* cerebellum (*Figure 4B*; n=3, p>0.5).

Disruption of 5' tRNA processing has previously been shown to stimulate translation of GCN4, the yeast equivalent of ATF4, although in a p-eIF2$\alpha$-independent manner (*Qiu et al., 2000*). To explore whether ATF4 activation may be induced by unprocessed *n-Tr20* generated from the B6J

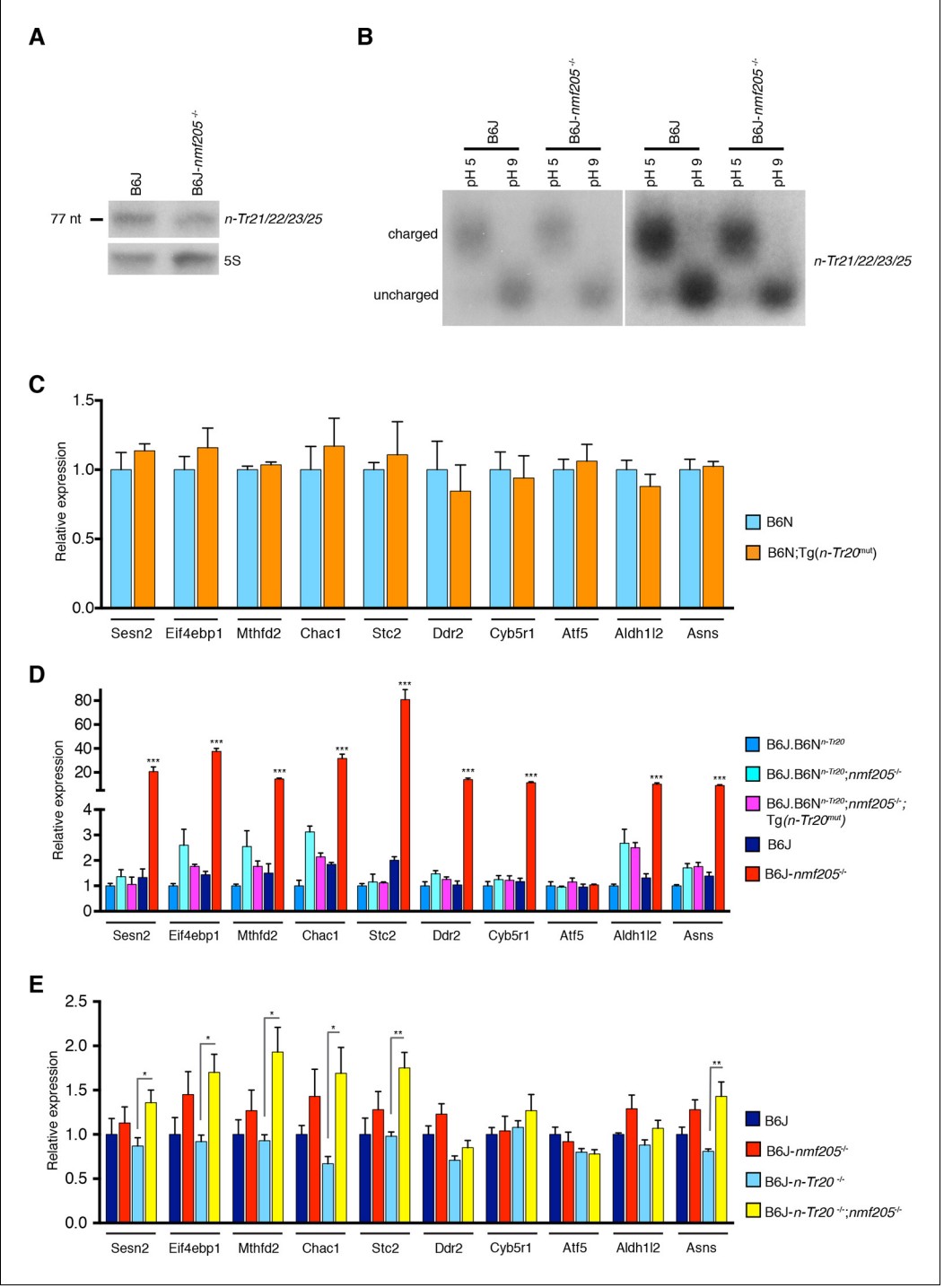

**Figure 4.** GCN2 activation in the B6J-*Gtpbp2*[nmf205-/-] brain is independent of uncharged tRNA[Arg]UCU and unprocessed *n-Tr20* tRNA. (**A**) Northern blot analysis of cerebellar RNA from 3-week-old B6J and B6J-*Gtpbp2*[nmf205-/-] (B6J-*nmf205*[-/-]) mice using pooled probes to *n-Tr21/22/23/25* tRNAs to assess the expression levels of the tRNA[Arg]UCU isodecoder family. 5S was used as internal control. (**B**) Charged (pH 5) and uncharged (pH 9) tRNA[Arg]UCU levels in the 3-week-old B6J and B6J-*Gtpbp2*[nmf205-/-] cerebellum. Note that the levels of uncharged tRNA[Arg]UCU are negligible in both the B6J-*Gtpbp2*[nmf205-/-] and B6J cerebellum. A short (right) and a longer (left) exposure are shown. (**C–E**) RT-qPCR analysis of expression of ATF4 target genes. (**C**) Overexpression of the mutant *n-Tr20* tRNA does not change expression of ATF4 targets in the 3-week-old cerebellum although increased levels of unprocessed *n-Tr20* are present. (n=3 mice per genotype) (**D**) Overexpression of the mutant *n-Tr20* tRNA does not change expression of ATF4 targets even in the *Gtpbp2*[nmf205-/-] mutant cerebellum at 3 weeks of age. (n=3) (**E**)

*Figure 4 continued on next page*

*Figure 4 continued*

ATF4 targets are significantly upregulated in the P0 brain of B6J-*n-Tr20*$^{-/-}$; *Gtpbp2*$^{nmf205-/-}$ mice, although no unprocessed or uncharged forms of *n-Tr20* are present. (n=4 mice per genotype) Error bars = SEM. *p<0.05, **p<0.01, and ***p<0.001 (Student's unpaired two-tailed *t* tests, C, E; one-way ANOVA, D).

The following figure supplements are available for figure 4:

**Figure supplement 1.** Generation of mice overexpressing mutant *n-Tr20*.

**Figure supplement 2.** Generation of mice with the deleted *n-Tr20* allele.

allele, we generated transgenic B6N mice that expressed the mutant *n-Tr20* gene (B6N-Tg-nTr20$^{mut}$) at levels 6-fold above that observed in B6J brain (*Figure 4—figure supplement 1*). Quantitative PCR using cerebellar cDNA from B6N and B6N-Tg-nTr20$^{mut}$ mice was performed for the ATF4 target genes that showed highest fold expression changes between the B6J and B6J-*Gtpbp2*$^{nmf205-/-}$ cerebellum. As shown in *Figure 4C*, overexpression of the unprocessed *n-Tr20* did not result in upregulation of these genes.

To determine if ATF4 activation was influenced by interaction between overexpression of the unprocessed tRNA and loss of *Gtpbp2*, we crossed B6N mice (wild type for *n-Tr20*) that transgenically overexpress mutant *n-Tr20* (B6N-Tg-n-Tr20$^{mut}$) mice to B6J.B6N$^{n-Tr20}$-*Gtpbp2*$^{nmf205-/-}$ mice (wildtype for *n-Tr20*, mutant for *Gtpbp2*). RT-qPCR for the ATF4-target genes analyzed above was performed on cerebella from 3-week-old B6J.B6N$^{n-Tr20}$-*Gtpbp2*$^{nmf205-/-}$ with, and without, the mutant transgene, and B6J.B6N$^{n-Tr20}$ (*n-Tr20* wild type), B6J (*n-Tr20* mutant) and B6J-*Gtpbp2*$^{nmf205-/-}$ (mutant for *n-Tr20*; *Gtpbp2*$^{-/-}$) mice. As expected, the expression of ATF4 target genes was dramatically upregulated in the B6J-*Gtpbp2*$^{nmf205-/-}$ cerebellum relative to the other strains. However, the expression of the mutant *n-Tr20* transgene, even in the presence of the *Gtpbp2* mutation, did not alter expression of ATF4 target genes (*Figure 4D*). Together, these results suggest that the induction of ATF4 target gene expression in B6J-*Gtpbp2*$^{nmf205-/-}$ is not due to expression of the unprocessed *n-Tr20* tRNA.

For further evidence that loss of mature *n-Tr20* causes GCN2 activation in the absence of *Gtpbp2*, we generated B6J mice in which the *n-Tr20* gene was deleted by homologous recombination (B6J-*n-Tr20*$^{-/-}$; *Figure 4—figure supplement 2*). Mice heterozygous or homozygous for this deletion were born at expected Mendelian ratios. However when these mice were crossed with *Gtpbp2*$^{nmf205-/-}$ mice, mice that were homozygous for both the *n-Tr20* deletion and the *nmf205* mutation died shortly after birth, demonstrating that residual levels of aminoacylated *n-Tr20* in the B6J brain allows postnatal survival of B6J-*Gtpbp2*$^{nmf205-/-}$ mice.

RT-qPCR for the 10 most highly induced ATF4-target genes were performed on cDNA generated from the brain of P0 B6J, B6J-*Gtpbp2*$^{nmf205-/-}$, B6J-*n-Tr20*$^{-/-}$, and B6J-*n-Tr20*$^{-/-}$; *Gtpbp2*$^{nmf205-/-}$ mice (*Figure 4E*). Overall, these genes were expressed at higher levels in the B6J-*Gtpbp2*$^{nmf205-/-}$ brain relative to that of B6J mice (p<0.002; Student's pairwise *t* test), however expression differences for individual genes were not significant between these genotypes. This result correlates with higher levels of mature *n-Tr20* in the brains of B6J P0 mice relative to brains of P30 B6J mice (*Ishimura et al., 2014*). However in the P0 B6J-*n-Tr20*$^{-/-}$; *Gtpbp2*$^{nmf205-/-}$ brain, 6 of these 10 genes were significantly upregulated relative to the B6J-*n-Tr20*$^{-/-}$ brain. These results demonstrate that complete loss of *n-Tr20* on the *Gtpbp2*-deficient background results in ATF4 activation.

## GCN2-ATF4 signaling pathway promotes survival for B6J-*Gtpbp2*$^{nmf205-/-}$ neurons

The effects of GCN2-ATF4 pathway on cell survival vary in an apparently context-dependent manner. To determine the effect of GCN2-ATF4 pathway on the progression of neurodegeneration induced by ribosome stalling, we compared locomotion and cerebellar pathology in B6J-*Gtpbp2*$^{nmf205-/-}$; *Gcn2*$^{-/-}$ and B6J-*Gtpbp2*$^{nmf205-/-}$ mice. As expected, B6J-*Gtpbp2*$^{nmf205-/-}$ mice exhibited mild ataxia and tremors at 45 days of age, and no motor defects were observed in B6J.*Gcn2*$^{-/-}$ or B6J mice. However, ataxia and tremors were much more profound in B6J-*Gtpbp2*$^{nmf205-/-}$; *Gcn2*$^{-/-}$ mice than in B6J-*Gtpbp2*$^{nmf205-/-}$ mice (data not shown). Consistent with the increased severity of the locomotor

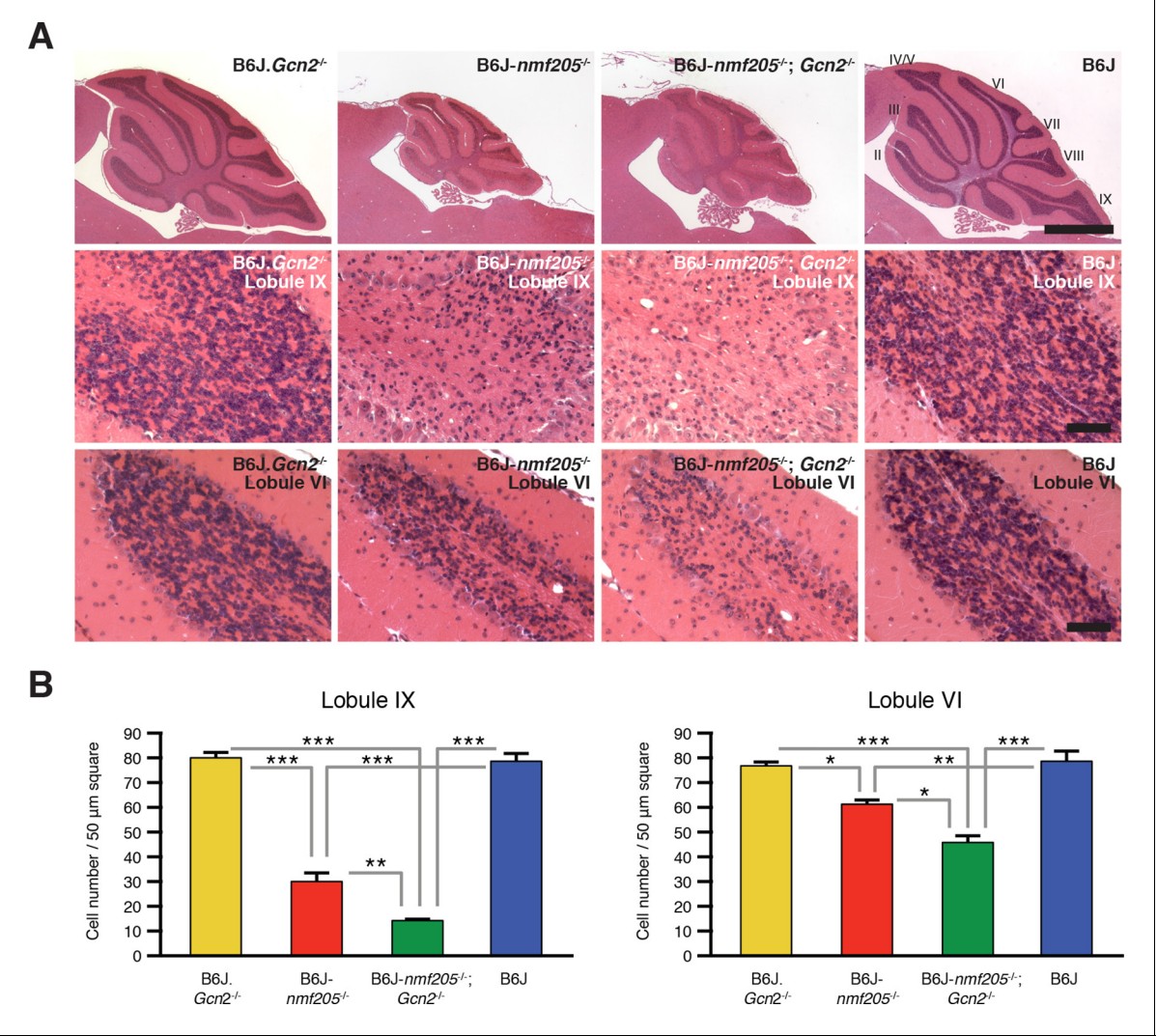

**Figure 5.** Loss of GCN2 accelerates B6J-*Gtpbp2^{nmf205-/-}* cerebellar degeneration. (**A**) Hematoxylin and eosin- stained sagittal sections of B6J.*Gcn2^{-/-}*, B6J-*Gtpbp2^{nmf205-/-}* (B6J-*nmf205^{-/-}*), B6J-*Gtpbp2^{nmf205-/-}*; *Gcn2^{-/-}* and B6J cerebella. Higher-magnification images of cerebellar lobule IX and lobule VI are shown on lower panels. Scale bars, 1 mm (top row), and 50 μm (bottom two rows). (**B**) Numbers of granule cells (n=3 mice/genotype). Means ± SEM are shown. *p<0.05, **p<0.01, and ***p< 0.001 (one-way ANOVA).

phenotype, histological examination showed an increase in loss of granule cells in the B6J-*Gtpbp2^{nmf205-/-}*; *Gcn2^{-/-}* cerebellum relative to the degeneration of these neurons in the B6J-*Gtpbp2^{nmf205-/-}* mice (*Figure 5*). No neuropathology was observed in B6J.*Gcn2^{-/-}* or B6J mice as expected. Furthermore, like B6J-*Gtpbp2^{nmf205-/-}* mice, neurodegeneration was not observed in the cerebellum of B6J-*Gtpbp2^{nmf205-/-}*; *Gcn2^{-/-}* mice at 3-weeks of age (data not shown), suggesting that the accelerated neurodegeneration in B6J-*Gtpbp2^{nmf205-/-}*; *Gcn2^{-/-}* was not due to a change in the onset of neurodegeneration but due to acceleration in progression of cerebellar granule cell loss.

In addition to accelerating cerebellar neuron loss, loss of GCN2 in B6J-*Gtpbp2^{nmf205-/-}* mice leads to novel sites of neuron loss. Degeneration of CA1 pyramidal neurons does not occur in B6J-*Gtpbp2^{nmf205-/-}* mice. However extensive death of these neurons was observed in the B6J-*Gtpbp2^{nmf205-/-}*; *Gcn2^{-/-}* hippocampus further supporting that activation of GCN2 is protective against neuron death in B6J-*Gtpbp2^{nmf205-/-}* mice (*Figure 6*).

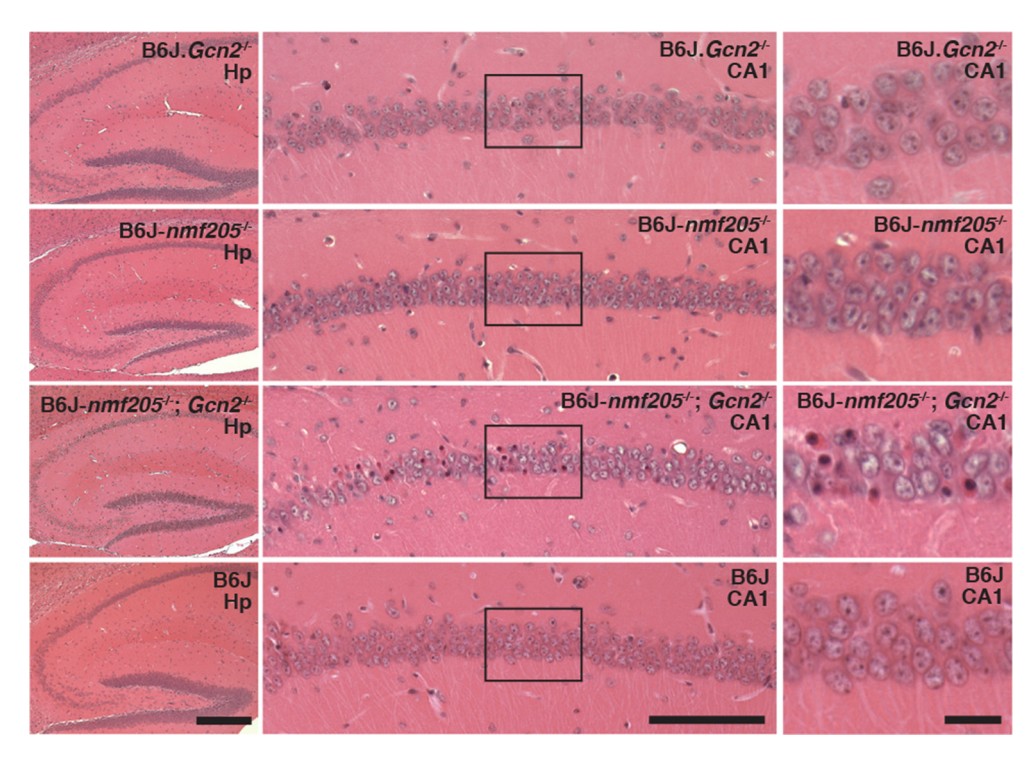

**Figure 6.** GCN2 protects against loss of B6J-*Gtpbp2*[nmf205-/-] CA1 hippocampal neurons. (**A**) Hematoxylin and eosin stained sagittal sections of 6-week-old B6J.*Gcn2*[-/-], B6J-*Gtpbp2*[nmf205-/-] (B6J-*nmf205*[-/-]), B6J- *Gtpbp2*[nmf205-/-]; *Gcn2*[-/-], and B6J hippocampi. Higher magnification images of hippocampal CA1 are also shown. Scale bars, 250 µm (left panels), 100 µm (middle panels), and 20 µm (right panels).

## Discussion

Failure to resolve stalled elongation complexes in mammalian neurons results in degeneration of these cells, but the signaling pathways evoked by these complexes are unknown. Here we show that GCN2 activates the integrated stress response in the brain of mice with stalled ribosomes, clearly demonstrating an important feedback loop between elongation defects, in particular ribosome stalling, and translation initiation. A recent study demonstrated that ribosomes bound to rare and thus slowly decoded codons near the initiation codon of yeast reporter genes prevent the binding of subsequent initiation complexes and decrease protein production (*Chu et al., 2014*). Our studies further connect translation elongation and translation initiation by showing that stalled ribosomes can be sensed by GCN2 leading to reprogramming of initiation.

Both mouse and yeast GCN2 are activated by deprivation of any amino acid or by conditions that mimic low amino acid availability such as aminoacyl-tRNA synthetase inhibitors and amino acid analogs, demonstrating that GCN2 is an important monitor of cellular amino acid levels. GCN2 is also activated by glucose deprivation, proteasome inhibition, and other conditions, many of which also decrease the amino acid pool, though in less obvious ways (*Castilho et al., 2014*; *Deng et al., 2002*; *Jiang and Wek, 2005*; *Yang et al., 2000*). A decrease in the intracellular amino acid pool in turn can lead to increased levels of deacylated tRNA. Competition assays have indicated that aminoacylated tRNA[Phe] binds the enzymatically inactive HisRS-like domain of GCN2 less efficiently than deacylated tRNA[Phe], suggesting GCN2 preferentially binds uncharged tRNAs (*Dong et al., 2000*). While not as efficient as acylated tRNAs, deacylated tRNA has been demonstrated to enter the A-site of the ribosome (*Murchie and Leader, 1978*) where models of GCN2 activation suggest that it is transferred to GCN2, possibly with the assistance of the GCN2 effector protein GCN1 to activate the kinase activity of GCN2 (*Marton et al., 1997*; *Ramirez et al., 1991*; *Wek et al., 1989*).

Interestingly, our results suggest that ribosome stalling, in the absence of changes in levels of deacylated tRNAs, is sufficient to activate GCN2. We examined tRNA$^{Arg}_{UCU}$ (the corresponding iso-acceptor for AGA, the only codon at which we observe stalled ribosomes upon ribosome profiling) and failed to detect changes in uncharged levels of *n-Tr20* isodecoders in B6J-*Gtpbp2$^{nmf205-/-}$* mutant cerebellum compared with that of B6J. Unprocessed tRNAs have also been shown to activate yeast ATF4, although this appears to occur independently of GCN2 (*Qiu et al., 2000*). Thus we tested if ATF4 activation is modulated by cerebellar overexpression of the unprocessed B6J-associated *n-Tr20* tRNA with or without the *Gtpbp2* mutation and found no change in ATF4 activation. Finally, ATF4 activation also occurs in the brains of B6J-*Gtpbp2$^{nmf205-/-}$* mice in which the *n-Tr20* is entirely deleted and thus no unprocessed tRNA is produced. These results raise the intriguing possibility that ribosome stalling may also contribute to GCN2 activation in other conditions, such as amino acid deprivation, in which like B6J-*Gtpbp2$^{nmf205-/-}$* neurons, lower levels of acylated tRNA are present.

The exact mechanism for GCN2 activation by ribosome stalling remains unclear. Yeast GCN2 loosely binds to ribosomes via its C-terminus and ribosome association is required for GCN2 activation during amino acid deprivation (*Narasimhan et al., 2004*; *Ramirez et al., 1991*; *Zhu and Wek, 1998*). Thus GCN2 activation may occur upon association with stalled ribosomes. However, multiple reports have failed to detect changes in the steady state ribosome association of yeast GCN2 during amino acid starvation (*Castilho et al., 2014*). Further, studies suggest that association of mouse GCN2 with ribosomes is lower, perhaps due to differences in the structures of yeast and mammalian GCN2 (*He et al., 2014*). In agreement, we failed to observe changes in the low levels of ribosome-associated GCN2 in the mouse brain of wild type or B6J-*Gtpbp2$^{nmf205-/-}$* mice (Ishimura, unpublished). As an alternative to alterations in ribosome binding, ribosome stalling may activate either cytoplasmic or ribosome-associated positive (e.g., GCN1 or GCN20) or inactivate negative (e.g., Yih1/IMPACT or eEF1A) regulators of GCN2 (*Castilho et al., 2014*). Lastly, ribosome stalling could alter cellular amino acid pools via unknown feedback mechanisms that in turn lead to GCN2 activation. Indeed, amino acid scarcity has been reported in cells during proteasome inhibition or after UV treatment, both of which have been shown to activate GCN2 (*Siegel and Swenson, 1964*; *Suraweera et al., 2012*).

Translational reprogramming mediated by the eIF2α kinases is thought to be adaptive during stress by transiently shutting protein synthesis to prevent waste of cellular resources until management of stress has occurred. However recent evidence suggests that under several stress conditions, phospho-eIF2α/ATF4 may facilitate apoptosis. For example, *Atf4*-deficiency reduced oxidative stress-induced neuron death and death of ER-stressed mouse embryonic fibroblasts and β-cells (*Han et al., 2013*; *Krokowski et al., 2013*; *Lange et al., 2008*). Knockdown of *Atf4* also attenuated neuronal loss induced by locally applied Aβ peptide (*Baleriola et al., 2014*). Recently, reports suggest that increased levels of eIF2α phosphorylation observed in transgenic mouse models of Alzheimer's disease and frontotemporal dementia and mice with prion disease amplifies neurodegeneration and defects in synaptic function (*Devi and Ohno, 2014*; *Ma et al., 2013*; *Moreno et al., 2012*; *Radford et al., 2015*; *Segev et al., 2015*), although other studies have failed to confirm some of these findings (*Devi and Ohno, 2013*; *Paesler et al., 2015*; *Sadleir et al., 2014*).

In contrast, we show that GCN2 activation protects neurons against death mediated by ribosome stalling. Deletion of *Gcn2* in B6J-*Gtpbp2$^{nmf205-/-}$* mice resulted in accelerated progression of cerebellar granule cell death. Furthermore in the hippocampus of B6J-*Gtpbp2$^{nmf205-/-}$*; *Gcn2$^{-/-}$*mice, we observed degeneration of CA1 pyramidal cells that is not observed in B6J-*Gtpbp2$^{nmf205-/-}$*, suggesting the eIF2α/ATF4 pathway is vital for survival of these neurons during ribosome stalling. Our findings are similar to those found by Harding et al in which induction of ATF4 downstream genes, particularly genes encoding various antioxidant enzymes, attenuated death of mouse embryonic fibroblast cells during amino acid starvation (*Harding et al., 2003*). Similarly this pathway was shown to be critical for survival of tumor cells under nutrient deprivation conditions and in cultured dopaminergic neurons against toxin-induced cell death. (*Sun et al., 2013*; *Ye et al., 2010*). These results suggest that the type of stress may influence the response of cells to reduction in initiation. Indeed, under mild amino acid stress conditions (two–five fold decrease in tRNAs) a decrease in initiation has been predicted to also decrease protein production via yeast whole-cell modeling simulations in which ribosomes have been suggested to be rate limiting for protein production (*Shah et al., 2013*).

However, under severe amino acid stress, reducing initiation is predicted to have the opposite effect and significantly increase protein production. This latter model suggests that under severe stress, cellular protein production is limited by elongation rather than initiation rates, and reducing the rate of initiation increases the pools of free ribosomes and free tRNAs that correspond to the depleted amino acid. While it is not clear if this model also applies to mammalian cells (and/or neurons) or other types of proteotoxic stress, such alternative consequences on protein production in different stress conditions may account for differential responses upon loss of the eIF2a/ATF pathway. It is tempting to suggest that protection by GCN2 is conferred by phosphorylation of eIF2α and the resulting decrease in the translational burden. However, recent findings have shown that cellular stress can cause downregulation of global translation during cellular stress independently of eIF2α phosphorylation (*Knutsen et al., 2015*), raising the possibility that eIF2α is not necessarily directly causative in ameliorating neuron death in the B6J- $Gtpbp2^{nmf205-/-}$ brain.

Loss of *Gcn2* in the B6J-$Gtpbp2^{nmf205-/-}$ cerebellum not only resulted in reduction of eIF2α phosphorylation and attenuated upregulation of ATF4 target genes, but also caused increased expression of genes that regulate nucleolar function, including those involved in ribosome biogenesis and processing of non-coding RNAs. Several of these genes were also upregulated, albeit at lower levels, in the B6J-$Gtpbp2^{nmf205-/-}$ cerebellum suggesting that neurons may sense a deficiency of functional ribosomes and correspondingly activate genes involved in ribosome biogenesis. The failure to decrease protein synthesis via GCN2-mediated eIF2α phosphorylation may result in a further depletion of functional ribosomes necessitating additional compensatory increases in expression of genes controlling nucleolar function. Interestingly, human mutations in genes that control ribosome biogenesis have been shown to induce nucleolar stress and a variety of pathologies (*Armistead and Triggs-Raine, 2014*; *Danilova and Gazda, 2015*; *Yelick and Trainor, 2015*). More experiments are necessary to test whether nucleolar stress contributes to death of neurons during conditions of ribosome stalling.

## Materials and methods

### Mouse strains

The *Gtpbp2^{nmf205-/-}* mutant strain was generated by ENU-treatment of C57BL/6J (B6J) mice as described previously (*Ishimura et al., 2014*). B6J.B6N$^{n-Tr20}$ mice and B6J.B6N$^{n-Tr20}$; *Gtpbp2^{nmf205-/-}* were described previously (*Ishimura et al., 2014*). B6J.*Gcn2^{-/-}* mice (*Eif2ak4^{tm1.2Dron}*) were obtained from The Jackson Laboratory. To generate Tg(n-Tr20$^{B6J}$) transgenic mice a 1.2-kb PCR fragment containing the mutant *n-Tr20* gene was amplified from B6J genomic DNA and injected into the pronuclei of B6N zygotes. Founders were identified by PCR using primers specific for the B6J *n-Tr20* allele (*mutant tRNA forward*, 5'ggacttctaatccagaggttgt3'; *common reverse*, 5'tatcccatcacgaagcaaaac3').

To generate a targeting construct for the deletion of the *n-Tr20* gene, a 251-bp DNA fragment consisting of the wild-type *n-Tr20* gene and 80-bp upstream and a 21-bp downstream genomic sequence followed by a loxP site was PCR amplified. This amplicon was cloned into a vector containing a loxP-FRT-neo-FRT cassette, downstream of the second FRT site. For recombineering the tRNA locus, this construct was amplified with overhangs and electoporated into *E. coli* SW105 cells carrying a BAC (RP24-205L18). The modified BAC was then linearized at BsiWI sites in the BAC backbone and electroporated into C57BL/6J ES cells. Southern blot and loss-of-allele (LOA) analysis were performed to identify clones with the targeted allele, and clones were injected into C57BL/6J *Perfect Host* blastocysts (*Taft et al., 2013*). For germline transmission of the mutant alleles, male chimeras were mated to C57BL/6J females. The floxed *n-Tr20* tRNA locus was deleted by mating heterozygous mice mice with B6J.*EIIa-Cre* mice obtained from The Jackson Laboratory. Deletion of the *n-Tr20* locus was confirmed by PCR analysis of tail genomic DNA using primers flanking the locus *forward 1*, 5'ggcgcgcctagtcgacataact3'; *forward 2*, 5'gcaggatgctgagatggctc3'; *common reverse*, 5'tatcccatcacgaagcaaaac3'). The Jackson Laboratory Animal Care and Use Committee approved all animal protocols.

## Microarray analysis

The cerebellum and hippocampus were collected from B6J and B6J- *Gtpbp2*[nmf205-/-] mice at 3 weeks and 5 weeks of age and RNA extraction was performed with Trizol (Invitrogen). cRNA was prepared from each sample and hybridized to Affymetrix Mouse Gene 1.0 ST Arrays using standard methods. Three biological replicates were performed for each time point and genotype. Average signal intensities for each probe set were calculated using Expression Console software (Version 1.1, Affymetrix) using the average (RMA) method. $F_s$, a modified F-statistic incorporating shrinkage estimates of variance components from within the R/-MAANOVA package was used to determine differentially expressed genes by one pairwise comparisons (*Cui et al., 2005*; *Wu et al., 2003*). Permutations analysis was used to calculate the levels of statistical significance of pairwise comparisons and these were adjusted for multiple testing using the false discovery rate, q-value method (*Storey, 2002*). Differentially expressed genes are declared at an FDR q-value threshold of 0.05.

## RNA-Seq

Library preparation for gene expression comparison among B6J, B6J-*Gcn2*[-/-], B6J- *Gtpbp2*[nmf205-/-], and B6J-*Gtpbp2*[nmf205-/-]; *Gcn2*[-/-] was performed based using Illumina TruSeq methodology. Cerebellum at 3 weeks of age from each genotype was isolated and immediately frozen in liquid nitrogen. Total RNA was extracted using Trizol reagent (Life Technologies). RNA were further purified using the Qiagen RNeasy MinElute Cleanup Kit per manufacturer's instructions (Qiagen) and the quality of the purified RNA was assessed using an Agilent 2100 Bioanalyzer instrument and RNA 6000 Nano LabChip assay. Messenger RNA was purified using biotin-tagged poly dT oligonucleotides and streptavidin-coated magnetic beads. After fragmentation of mRNA, libraries were prepared using TruSeq RNA Sample Prep Kit v2 (Illumina). Quality of libraries was assessed on Agilent 2100 Bioanalyzer. Paired-end reads (2X100bp) were obtained using the HiSeq2500 (Illumina). The sequence reads were processed to a set of comprehensive QC metrics and then FASTQ data file of nucleotide sequences.

Each library was split into two lanes for sequencing, yielding two FASTQ files that were then merged. Read mapping and quality control, read clipping and trimming were performed at the command line with options: 1) fastx_clipper -Q33 -a CTGTAGGCACCATCAAT -l 25 -c -n -v -i $fastqfile -o temp_clipped.fastq and 2) fastx_trimmer -Q33 -f 2 -l 100 -i temp_clipped.fastq -o temp_trimmed.fastq. The resulting reads were mapped to a fasta file containing all coding transcripts plus the last 18nts of the 5'UTR and the first 18nts of the 3'UTR downloaded from Ensembl Biomart (http://www.ensembl.org/biomart/martview/) using bowtie: bowtie -S -p 16 -n 1 -m 1 -l 23 –norc $fastafile temp_trimmed.fastq where only reads uniquely matching to the sense strand and containing not more than one mismatch were considered.

After mapping, read counts were calculated with an ad-hoc script, and then genes with significant up and down-regulation were calculated using DESeq2 (*Love et al., 2014*). We applied DESeq2 to detect differentially expressed genes at the default stringency, i.e. p-value of 0.1 (multiple-testing adjusted) or lower. Genes with low read counts were removed from the analysis using the independent filtering option in DESeq2 as determined by the p-value=0.1 threshold. We note that this results in genes with low read counts being given an adjusted p value of NA, and therefore they are not plotted in our volcano plots.

## Gene ontology

Upstream regulator analysis was performed with IPA (Ingenuity Pathway Analysis) browser (http://pages.ingenuity.com/Ingenuity_Login.html). Gene Ontology (GO) and Kegg pathway analysis was performed using the DAVID bioinformatics web server (http://david.abcc.ncifcrf.gov/) by uploading the gene lists from microarray analysis that were significantly upregulated or downregulated in B6J-*Gtpbp2*[nmf205-/-] compared to B6J brains or gene lists from RNAseq analysis (*Huang da et al., 2009*). The functional annotation chart and clustering analysis modules were employed for gene-term enrichment analysis. GO terms and Kegg pathway terms with a p-value < 0.05 were considered enriched.

Activated microglia and astrocyte gene lists were obtained from Glia Open Access Database (GOAD, http://bioinf.nl:8080/GOAD/databaseSelectServlet) (*Holtman et al., 2015*). Microglia related genes were obtained from the studies of Alzheimer's disease model (APP[Swe]; PS1dE9)

(*Orre et al., 2014*) and of ALS disease model (Sod1, Endpoint) (*Chiu et al., 2013*). Astrocyte related genes were obtained from the study of Alzheimer's disease model (App-Ps1) (*Orre et al., 2014*). Differentially expressed >1.5 fold genes with a p value < 0.05 were analyzed.

## Histology and immunofluorescence

Mice were transcardically perfused with Bouin's fixative and brains were embedded in paraffin. Sections were stained with hematoxylin and eosin according to standard protocols. For immunofluorescence, mice were perfused with 4% paraformaldehyde (PFA) and brains were paraffin embedded. Sections were stained with rabbit anti-Iba1 (Wako, Richmond, VA) or a mouse anti-GFAP (Sigma Aldrich, St. Louis, MO). Signal was visualized with Alexa Fluor 488-congugated or 555-conjugated donkey secondary antibodies (Invitrogen).

## Cell counts

Granule cells, identified by their distinct nuclei were counted in a $50 \times 50$ μm area from lobule VI and IX in hematoxylin and eosin stained sections that were taken at midline from three mice of each genotype.

## Northern blot and tRNA aminoacylation analysis

RNA extraction, Northern blot analysis, and tRNA aminoacylation analysis were performed as described previously (*Ishimura et al., 2014*). Blots were hybridized with a $^{32}$P-labeled oligo probe (TCT_nonchromosome1; [*Ishimura et al., 2014*]) recognizing the common sequence among tRNA$^{Arg}_{UCU}$ isodecoders *n-Tr21, n-Tr22, n-Tr23*, and *n-Tr25*. Northern blot analysis of brain RNA from the B6J-*n-Tr20*$^{-/-}$ mice and B6N-Tg(n-Tr20$^{B6J}$) mice was performed using a $^{32}$P-labeled oligo probe specific to *n-Tr20* as described previously (*Ishimura et al., 2014*).

## Western blotting

Proteins were extracted from cerebellum at three weeks of age as described previously (*Carnevalli et al., 2004*) and were resolved on SDS-PAGE gels prior to transfer to PVDF membranes. Membranes were blocked with 5% bovine serum albumin prior to incubations with primary antibodies. Total eIF2α and phosphorylated eIF2α (Ser51) were detected using anti-human rabbit polyclonal antibodies (Cell Signaling Technology, Danvars, MA) and anti-human rabbit polyclonal antibody (Invitrogen), respectively. Glyceraldehyde-3-phosphate dehydrogenase (GAPDH) was detected using anti-human mouse monoclonal antibody (clone 3C2, Sigma Aldrich). Membranes were subsequently incubated with HRP-conjugated anti-rabbit or anti-mouse IgG (1:5,000; Bio-Rad Life Science, Hercules, CA). Signals were detected with ECL Plus (GE Healthcare Life Sciences) and band intensities were quantified using ImageJ software (NIH).

## RT-qPCR and *Xbp1* analysis

Cerebella were isolated and immediately frozen in liquid nitrogen. Total RNA was extracted with Trizol reagent (Life Technologies). 1.3 μg of total RNA was treated with DNAse I (Sigma) and then used for cDNA synthesis with oligo(dT)$_{20}$ primers and the SuperScript III First-Strand Synthesis System (Invitrogen). PCR were performed using iQ SYBR Green Supermix (Bio-Rad) using a iQ5 Multicolor Real-Time PCR Detection System (Bio-Rad). Primers were as follows: *Sesn2*-forward, 5'ccttctccacacccagacat3'; *Sesn2*-reverse, 5'agcctctggatcagcgagta3'; *Mthfd2*-forward, 5'ccgccagtcactcctatgtt3'; *Mthfd2*-reverse, 5'ggaggccatctacgttctca3'; *Ddr2*-forward, 5'tgatggagctgttgggtaca3'; *Ddr2*-reverse, 5'ccgttggtagcactttcgtt3'; *Aldh1l2*-forward, 5'gaggagcttctgccatcaac3'; *Aldh1l2*-reverse, 5'gcgagtccactgtgtcattg3'; *Chac1*-forward, 5'gtacggctccctagtgtgga3'; *Chac1*-reverse, 5'tcttcaaggagggtcaccac3', *Asns*-forward, 5'aagatgggtttctggctgtg3'; *Asns*-reverse, 5'gcaactttgccatttggttt3'; *Stc2*-forward, 5'gattcatttcaaggatctcc3'; *Stc2*-reverse, 5'ggttcacaaggtccacatag3'; *Atf5*-forward, 5'cctgtggattaaagggggta3'; *Atf5*-reverse, 5'agcgtggaagattgttcagc3'; *Eif4ebp1*-forward, 5'ctagccctaccagcgatgag3'; *Eif4ebp1*-reverse, 5'cttgggggacatagaagcat3'; *Cyb5r1*-forward, 5'gctttctgcttttgccaac3'; *Cyb5r1*-reverse, 5'caaagcccttgctgtaggtc3' *Gapdh*-forward, 5'cattgtcataccaggaaatg3', *Gapdh*-reverse, ggagaaacctgccaagtatg3'. Expression of each gene of interest is normalized to *Gapdh* using the 2$^{-\Delta\Delta CT}$ method (*Livek and Schimittgen, 2001*). Primers used for *Xbp1* splicing analysis were *Xbp1*-forward, 5'gaaccaggagttaagaacacg3', *Xbp1*-reverse, 5'aggcaacagtgtcagagtcc3'.

## Statistics

All data are presented as means ± SEM. Data were analyzed by one-way ANOVA or Student's t-test. $p < 0.05$ was considered statistically significant.

## Acknowledgements

We thank The Jackson Laboratory Cell Biology and Microinjection services, Genome Technologies, and the Mouse Mutant Repository, Tim Sterns for microarray analysis, Drs. Vivek Philip and Mridu Kapur for assistance with data analysis, Audrey Crane for ER-stress analysis, and Krystal Leigh Brown for technical assistance with mice. This work was supported by an NIH grant to SLA (NS094637) and core services were supported by CORE grant (CA34196). JHC was supported by NIH grant HG007554. SLA is an investigator of the Howard Hughes Medical Institute.

## Additional information

### Competing interests

SLA: Reviewing editor, *eLife*. The other authors declare that no competing interests exist.

### Funding

| Funder | Grant reference number | Author |
| --- | --- | --- |
| National Institute of Neurological Disorders and Stroke | NS094637 | Susan L Ackerman |
| Howard Hughes Medical Institute | | Susan L Ackerman |
| National Human Genome Research Institute | HG007554 | Jeffrey H Chuang |

The funders had no role in study design, data collection and interpretation, or the decision to submit the work for publication.

### Author contributions

RI, GN, Conception and design, Acquisition of data, Analysis and interpretation of data, Drafting or revising the article; ID, JHC, Acquisition of data, Analysis and interpretation of data; SLA, Conception and design, Analysis and interpretation of data, Drafting or revising the article

### Author ORCIDs

Susan L Ackerman, http://orcid.org/0000-0002-6740-593X

### Ethics

Animal experimentation: This study was performed in strict accordance with the recommendations in the Guide for the Care and Use of Laboratory Animals of the National Institutes of Health. All of the animals were handled according to approved institutional animal care and use protocol (TJL99055) at The Jackson Laboratory.

## Additional files

### Major datasets

The following dataset was generated:

| Author(s) | Year | Dataset title | Dataset URL | Database, license, and accessibility information |
|---|---|---|---|---|
| Ishimura R, Nagy G, Dotu I, Chuang J, Ackerman S | 2016 | Activation of GCN2 Kinase by Ribosome Stalling Links Translation Elongation with Translation Initiation | http://www.ncbi.nlm.nih.gov/geo/query/acc.cgi?acc=GSE79930 | Publicly available at the NCBI Gene Expression Omnibus (accession no: GSE79930). |

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
