## [Decision Letter]

Thank you for submitting your work entitled "Activation of GCN2 by Ribosome Stalling Links Translation Elongation with Translation Initiation" for consideration by *eLife*. Your article has been favorably evaluated by a Senior editor and three reviewers, one of whom is a member of our Board of Reviewing Editors.

The reviewers have discussed the reviews with one another and the Reviewing Editor has drafted this decision to help you prepare a revised submission.

Summary:

All three reviewers recognized the significance of your paper's conclusions in regards to GCN2 activation by mechanism(s) that do not entail uncharged tRNAs and that ribosome stalling activates the ISR via GCN2 to limit neurodegeneration in a mouse model of ribosome stalling. Their views diverged initially in regards to the extent that the genetic arguments put forth in support of the conclusion were sufficient. However, in the consultation that followed the reviewers agreed that a suitably revised paper would be acceptable for publication even without further mechanistic insight into the links between ribosome stalling and GCN2 activation.

Essential revisions:

1) Given that the papers central conclusions rest on analysis of an imaginatively curated collection of mutant mice, it is important to qualify the conclusions and consider alternative mechanisms by which the combination of the rare allele of tRNA_ARG and the deficiency in GTPBP2 might lead indirectly to canonical activation of GCN2. Of particular note are linkages between GCN2 and stresses that interfere with proteasomes. This could involve recycling of amino acids, and another means of indirect activation of GCN2.

2) As the role of eIF2α phosphorylation in limiting neurodegeneration is inferred from the phenotype of the GCN2 knockout (but not confirmed by further genetic manipulation of levels of eIF2α phosphorylation), the conclusion relating to the protective role of the ISR should be qualified.

3) Figure 3 shows that the effect of GCN2 deletion on levels of eIF2α phosphorylation is more evident in the B6J reference background than in the B6J-*nmf205^-/-^* background. This raises several questions: (a) Why then did the mRNA expression profiling show no differences between these two genotypes? (b) Figure 3 suggests involvement of other eIF2α kinases in the B6J-*nmf205^-/-^*. Has the status of the UPR been examined experimentally in the B6J-*nmf205^-/^* mice (XBP1 splicing, BiP expression)? The latter concern is further accentuated by the finding that EIF2AK3/PERK mRNA is elevated in the mutant mice (Figure 1).

4) Figure 4 are central to the paper's conclusion that tRNA uncharging does not correlate with GCN2 activation in the mutant mice. It would important to provide careful quantification (including shorter exposures) and statistical analysis of this key experiment. Extending the study to older mice (5 weeks old) would be very helpful.

---

## [Author Response]

Essential revisions:

1) Given that the papers central conclusions rest on analysis of an imaginatively curated collection of mutant mice, it is important to qualify the conclusions and consider alternative mechanisms by which the combination of the rare allele of tRNA_ARG and the deficiency in GTPBP2 might lead indirectly to canonical activation of GCN2. Of particular note are linkages between GCN2 and stresses that interfere with proteasomes. This could involve recycling of amino acids, and another means of indirect activation of GCN2.

The reviewers asked us to qualify our conclusions with alternative mechanisms by which GCN2 can be activated in B6J-*Gtpbp2^-/-^* mice. We have expanded on the reviewer’s suggestion of proteasome inhibition/GCN2 activation and amino acid pools in the Discussion.

2) As the role of eIF2α phosphorylation in limiting neurodegeneration is inferred from the phenotype of the GCN2 knockout (but not confirmed by further genetic manipulation of levels of eIF2α phosphorylation), the conclusion relating to the protective role of the ISR should be qualified.

The reviewers asked us to qualify our discussion of the protective role of the ISR, given that this is inferred from our analysis of B6J-*Gcn2-/-; Gtpbp2-/-*mice. We agree that experiments using further genetic or other manipulations of eIF2α phosphorylation levels will be needed in the future to unequivocally establish causality and have modified the Discussion accordingly.

3) Figure 3 shows that the effect of GCN2 deletion on levels of eIF2α phosphorylation is more evident in the B6J reference background than in the B6J-nmf205^-/-^ background. This raises several questions: (a) Why then did the mRNA expression profiling show no differences between these two genotypes?

The reviewers point out that *Gcn2* deletion decreases levels of eIF2α phosphorylation on the B6J background, but gene expression profiles show no differences between the B6J and B6J-*Gcn2^-/-^*cerebellum. To re-analyze relative expression levels with higher precision, we have performed RT-qPCR of the top 10 most upregulated genes in B6J-*nmf205*^-/-^ cerebellum. We found that for several genes the expression in B6J-*Gcn2*^-/-^ cerebellum is slightly, but significantly, lower than in the B6J cerebellum. Therefore, it is suggestive that deletion of *Gcn2* in B6J mice results in gene expression changes on a magnitude that is not readily detectable by RNA-Seq. These results are shown in a new figure, Figure 3—figure supplement 4.

(b) Figure 3 suggests involvement of other eIF2α kinases in the B6J-nmf205^-/-^. Has the status of the UPR been examined experimentally in the B6J-nmf205^-/^ mice (XBP1 splicing, BiP expression)? The latter concern is further accentuated by the finding that EIF2AK3/PERK mRNA is elevated in the mutant mice (Figure 1).

In Figure 1, IPA analysis detected enrichment of the EIF2AK3 (PERK) pathway. Many of these targets overlap with other eIF2α kinases and thus, detection of PERK does not exclude other activation of these other enzymes (in fact IPA detected enrichment of both PERK and GCN2 in IPA analysis of the downregulated genes in the B6J-*nmf205*^-/-^, *GCN2*^-/-^ cerebellum). However, we now include negative data from our analysis of UPR activation in the B6J-*nmf205*^-/-^ cerebellum (see text and Figure 3—figure supplement 4).

4) Figure 4 are central to the paper's conclusion that tRNA uncharging does not correlate with GCN2 activation in the mutant mice. It would important to provide careful quantification (including shorter exposures) and statistical analysis of this key experiment. Extending the study to older mice (5 weeks old) would be very helpful.

The reviewers suggested we quantitate tRNA levels and charged/uncharged tRNA levels. TRNA levels were decreased in the B6J-*nmf205*^-/-^ cerebellum as described in the text. However, no difference in the ratio of uncharged to charged tRNA was detected between genotypes. We have included shorter exposures of these blots. These experiments were all done on 3-week-old and not 5-week-old mice to avoid artifacts due to cell death.